# The Impact of the Hippo Pathway and Cell Metabolism on Pathological Complete Response in Locally Advanced Her2+ Breast Cancer: The TRISKELE Multicenter Prospective Study

**DOI:** 10.3390/cancers14194835

**Published:** 2022-10-03

**Authors:** Eriseld Krasniqi, Francesca Sofia Di Lisa, Anna Di Benedetto, Maddalena Barba, Laura Pizzuti, Lorena Filomeno, Cristiana Ercolani, Nicola Tinari, Antonino Grassadonia, Daniele Santini, Mauro Minelli, Filippo Montemurro, Maria Agnese Fabbri, Marco Mazzotta, Teresa Gamucci, Giuliana D’Auria, Claudio Botti, Fabio Pelle, Flavia Cavicchi, Sonia Cappelli, Federico Cappuzzo, Giuseppe Sanguineti, Silverio Tomao, Andrea Botticelli, Paolo Marchetti, Marcello Maugeri-Saccà, Ruggero De Maria, Gennaro Ciliberto, Francesca Sperati, Patrizia Vici

**Affiliations:** 1Division of Medical Oncology 2, IRCCS Regina Elena National Cancer Institute, 00144 Rome, Italy; 2Phase IV Clinical Studies Unit, IRCCS Regina Elena National Cancer Institute, 00144 Rome, Italy; 3Pathology Unit, IRCCS Regina Elena National Cancer Institute, 00144 Rome, Italy; 4Department of Medical, Oral and Biotechnological Sciences, Center for Advanced Studies and Technology (CAST), G. D’Annunzio University, 66100 Chieti, Italy; 5Department of Innovative Technologies in Medicine and Dentistry, Centre for Advanced Studies and Technology (CAST), G. D’Annunzio University, 66100 Chieti, Italy; 6“Sapienza” University of Rome, Polo Pontino, 04011 Aprilia, Italy; 7Division of Oncology, San Giovanni Hospital, 00184 Rome, Italy; 8Breast Unit, Candiolo Cancer Institute, Fondazione del Piemonte per l’Oncologia-IRCCS (Istituti di Ricovero e Cura a Carattere Scientifico), 10060 Candiolo, Italy; 9Medical Oncology Unit, Belcolle Hospital, 01100 Viterbo, Italy; 10Medical Oncology, Sandro Pertini Hospital, 00157 Rome, Italy; 11Department of Surgery, IRCCS Regina Elena National Cancer Institute, 00144 Rome, Italy; 12Department of Radiation Oncology, IRCCS Regina Elena National Cancer Institute, 00144 Rome, Italy; 13Department of Radiological, Oncological and Anatomo-Pathological Sciences, “Sapienza” University of Rome, 00185 Rome, Italy; 14Istituto Dermopatico dell’Immacolata, IDI-IRCCS, 00167 Rome, Italy; 15Clinical Trial Center, Biostatistics and Bioinformatics, IRCCS Regina Elena National Cancer Institute, 00144 Rome, Italy; 16Department of Translational Medicine and Surgery, Università Cattolica del Sacro Cuore, 00168 Rome, Italy; 17Fondazione Policlinico Universitario “A. Gemelli”, IRCCS (Istituti di Ricovero e Cura a Carattere Scientifico), 00168 Rome, Italy; 18Scientific Direction, IRCCS Regina Elena National Cancer Institute, 00144 Rome, Italy; 19Clinical Trial Center, Biostatistics and Bioinformatics, San Gallicano Dermatological Institute IRCCS, 00144 Rome, Italy

**Keywords:** Hippo pathway, cell metabolism, Her2−positive breast cancer, pathological complete response

## Abstract

**Simple Summary:**

Pathological complete response (pCR) is a key outcome in locally advanced Her2+ breast cancer (BC) patients treated with anti-Her2−based neoadjuvant therapy. Several clinical and biological features may affect pCR rate, but reliable predictors are needed for clinical practice. The Hippo pathway and its main transducers, Yes-associated protein (YAP) and transcriptional coactivator with PDZ-binding motif (TAZ), play a relevant role in treatment outcomes. We herein present evidence concerning the impact of the immunohistochemical expression of key Hippo transducers and their regulators, namely AMP-activated protein kinase (AMPK), Stearoyl-CoA-desaturase 1 (SCD1), and HMG-CoA reductase (HMGCR), on the pCR rate of 65 Her2+ BC patients receiving neoadjuvant trastuzumab-based therapy. Low expression of TAZ, especially if concomitant with low expression of YAP and HMGCR and high expression of SCD1, was a negative predictor of pCR, although not confirmed in the multivariate analysis. However, our findings were concordant with overall survival data from the TCGA cohort.

**Abstract:**

The Hippo pathway and its two key effectors, Yes-associated protein (YAP) and transcriptional coactivator with PDZ-binding motif (TAZ), are consistently altered in breast cancer. Pivotal regulators of cell metabolism such as the AMP-activated protein kinase (AMPK), Stearoyl-CoA-desaturase 1 (SCD1), and HMG-CoA reductase (HMGCR) are relevant modulators of TAZ/YAP activity. In this prospective study, we measured the tumor expression of TAZ, YAP, AMPK, SCD1, and HMGCR by immunohistochemistry in 65 Her2+ breast cancer patients who underwent trastuzumab-based neoadjuvant treatment. The aim of the study was to assess the impact of the immunohistochemical expression of the Hippo pathway transducers and cell metabolism regulators on pathological complete response. Low expression of cytoplasmic TAZ, both alone and in the context of a composite signature identified by machine learning including also low nuclear levels of YAP and HMGCR and high cytoplasmic levels of SCD1, was a predictor of residual disease in the univariate logistic regression. This finding was not confirmed in the multivariate model including estrogen receptor > 70% and body mass index > 20. However, our findings were concordant with overall survival data from the TCGA cohort. Our results, possibly affected by the relatively small sample size of this study population, deserve further investigation in adequately sized, ad hoc prospective studies.

## 1. Introduction

The Hippo signaling pathway is an evolutionarily conserved cascade of serine/threonine protein kinases and transcription co-factors with a major role in organ growth, tissue regeneration, and immune regulation [1,2]. Its deregulation has been consistently demonstrated across many tumors, including breast cancer, acting mainly like a tumor-suppressing factor [3]. More recent evidence supports a cancer type-dependent function of the Hippo pathway in tumorigenesis, but the mechanism is not fully understood [4,5,6]. Stress and an unfavorable environment for tissue growth represent the main activation triggers for the Hippo pathway in normal conditions. The core components of the Hippo pathway include two key effectors, Yes-associated protein (YAP) and transcriptional coactivator with PDZ-binding motif (TAZ) [7,8,9,10,11,12,13,14,15,16]. A scheme of the Hippo pathway machinery and the related signaling hubs is represented in Figure 1.

In response to stress, a cascade of Hippo pathway kinases is unleashed, leading to the activation of the Large Tumor Suppressor Kinases 1 and 2 (LATS1/2), which in turn exert inhibition on YAP and TAZ by reversibly phosphorylating them [17,18,19,20]. Phosphorylation causes cytoplasmic retention of YAP and TAZ, making the two proteins ubiquitination and proteasomal degradation targets [19,20,21,22]. Conversely, unphosphorylated YAP and TAZ enter the nucleus and combine with transcription factors that have DNA binding domains, such as TEA domain proteins (TEAD1, TEAD2, TEAD3, and TEAD4) and the RUNX family [23,24,25]. In this manner, YAP and TAZ regulate the gene expression of target genes related to cell growth and organ development. The abnormal modulation of the Hippo pathway, as it occurs in cancer development, makes it a key element in cancer cell invasion and migration by promoting the epithelial–mesenchymal transition (EMT) [26,27,28]. Besides the regulation inside the Hippo pathway, some additional key actors involved in cell metabolism can directly or indirectly modulate TAZ and YAP activity. For instance, AMP-activated protein kinase (AMPK), which is a key cellular energy sensor that monitors ATP/AMP levels, can phosphorylate and inhibit YAP when glucose levels are low, shifting cellular metabolism from a synthesis-replicative mode into an energy-generating state by favoring nutrient uptake and breakdown. Conversely, when cellular glucose and ATP levels increase, the AMPK inhibition ceases, favoring TAZ/YAP activation, which stimulates cell growth and replication [29,30]. Similarly, the cellular abundance of important structural lipids such as monounsaturated fatty acids (MFAs) and Acetyl-CoA fosters cell growth and replication by inciting TAZ/YAP activity with distinct mechanisms. Stearoyl-CoA-desaturase 1 (SCD1) is a pivotal enzyme in the process of endogenous MFA synthesis. Pre-clinical data showed that SCD1 inhibition and the consequent reduced concentration of cellular MFAs determine a reduced activity of TAZ/YAP, possibly by reducing homologous wingless (wg) and Int-1 (Wnt) signaling and β-catenin activity, which normally opposes their proteasomal degradation [31]. On the other hand, the Mevalonate pathway has the important role of converting Acetyl-CoA into Farnesyl-PP, which is the common predecessor of important cellular building blocks such as cholesterol, ubiquinones, Heme A, sterols, and geranylgeranyl diphosphate needed for protein prenylation. A critical enzyme that modulates the activity of this pathway is represented by HMG-CoA reductase (HMGCR). Studies have shown that HMGCR inhibition through statins suppresses YAP/TAZ nuclear translocation, possibly through the inhibition of Ras Homolog Family Member A (RhoA) geranylgeranylation, which blocks LATS1/2 inhibition (Figure 1) [32,33,34,35]. Functional impairment of the Hippo pathway in human cancers is relatively frequent, yet somatic mutations and copy number alterations are relatively uncommon in this pathway [36]. Nonetheless, the disruption of the Hippo pathway might result from the cross-interaction with other perturbed molecular pathways, which will equally lead to an altered expression of TAZ/YAP, as observed in a wide spectrum of human cancers [23]. Studies have shown that the Hippo transducer TAZ has a relevant role in breast cancer (BC), conferring its cells cancer stem cell-related traits and being required for their metastatic activity and chemoresistance [37,38,39].

Within a dedicated research pipeline, we previously addressed the role played by this regulatory pathway in BC and other tumors, along with the impact of its deregulation on key treatment outcomes, with encouraging results concerning the prognostic and/or predictive potentials of the biomarkers of interest [40,41,42,43,44,45]. In specific reference to Her2+ BC, in a prior retrospective study, we investigated the association of TAZ with pathologic complete response (pCR) in 61 Her2+ BC patients following exposure to neoadjuvant chemotherapy plus trastuzumab. A significant negative correlation between high TAZ expression and pCR emerged in Her2+ Luminal B tumors with high expression of estrogen and progesterone receptors (ER and PR, respectively) [46]. Moreover, in the BC clinical setting, TAZ expression was positively correlated with Her2 positivity and negatively correlated with disease-free survival [38].

With this awareness of the limitations stemming from the retrospective nature of our prior study in Her2+ locally advanced BC and in light of the previously cited evidence from pre-clinical and clinical studies carried out by other research groups and ours, we designed this multicentric, prospective study to investigate the impact of key components of the Hippo pathway on pCR in locally advanced BC patients treated with trastuzumab-based neoadjuvant regimens. Given the tight connection between the pathway of interest and cell metabolism modulation, in this same study population, we also explored the impact of selected cell metabolism regulators’ determinants on pCR.

## 2. Methods

### 2.1. Study Participants and Procedures

This is a multicenter, prospective observational study conceived to investigate the potential of specific biomarkers in predicting pCR in Her2+ BC patients treated with neoadjuvant trastuzumab-based regimens. The study was carried out in full accordance with the guidelines for Good Clinical Practice and the Declaration of Helsinki. The inherent protocol and consent form were approved by the Institutional Review Board (IRB) of the coordinating and satellite enrolling centers. Overall, eight cancer centers adhered to our study. Enrolment was performed between August 2014 and March 2017. Written informed consent was obtained from all patients. Patients were deemed suitable for inclusion if diagnosed with histologically proven, locally advanced Her2+ BC (stages II/III) and treated with trastuzumab-based neoadjuvant therapy and subsequent surgery. Additional conditions to be eligible for study inclusion were the availability of complete data relative to treatment outcomes, finalization of programmed pre-operatory treatment, and presence of a sufficient amount of biological material in baseline tumors’ biopsies for the immunohistochemical (IHC) assessment of the biomarkers of interest.

As previously mentioned, the neoadjuvant treatment schedule contained trastuzumab for all patients. The following regimens were in use at the time of enrollment and were adopted in the study design: anthracycline plus cyclophosphamide for four cycles, followed by paclitaxel or docetaxel plus trastuzumab; the same schedule with the administration of trastuzumab for the whole duration of chemotherapy; and the reverse sequence schedule starting with taxane followed by an anthracycline-containing regimen, with the administration of trastuzumab for the whole duration of chemotherapy.

This study primarily aimed to assess whether, at the tumor tissue level, the IHC expression of key Hippo pathway transducers (TAZ and YAP) and key cellular metabolism regulators (AMPK, HMGCR, and SCD1), alone or in combination with each other, influenced pCR at surgery. We also sought to identify further clinical and/or pathological determinants of pCR and explored associations between key patient and disease characteristics and IHC expression of the biomarkers of interest.

Pathological complete response (pCR) was defined as the absence of residual invasive disease at surgery in both breast and axillary lymph nodes (ypT0/is ypN0) and was assessed by expert pathologists at each participating cancer center. Three-micrometer sections of formalin-fixed paraffin-embedded BC tissue were cut on SuperFrost Plus slides (Menzel-Gläser, Braunschweig, Germany). Estrogen and progesterone receptors were assessed using the monoclonal antibodies (MoAbs) 6F11 and 1A6 (Leica Biosystems, Wetzlar, Germany), respectively. Her2 and Ki67 were measured using the polyclonal antibody (PoAb) A0485 (Agilent, Santa Clara, CA, USA) and the MoAb MIB-1 (Agilent), respectively. TAZ, YAP, AMPK, HMGCR, and SCD1 protein expression was quantified by using the MoAb anti-TAZ (M2-616, BD Pharmingen, San Diego, CA, USA), the MoAb anti-YAP (H-9, Santa Cruz, Dallas, TX, USA), the PoAb anti-AMPK alpha1 (phospho T172+T183, Abcam, Cambridge, UK), the PoAb anti-HMGCR (Sigma-Aldrich, St. Louis, MO, USA), and the MoAb anti-SCD1 (CD.E10, Abcam), respectively. The following dilution for each antibody was used: anti-TAZ clone (M2-616, BD Pharmingen), 1:400, pH 6; anti-YAP (H-9, sc-271134, Santa Cruz), 1:200, pH 8; anti-AMPK alpha 1 (phospho T172+T183 antibody, Abcam), 1:300, pH 6; anti-SCD1 antibody (CD.E10, Abcam), 1:300, pH 8; and anti-HMGCR polyclonal antibody (Sigma), 1:150, pH 8. Immunoreactions were revealed in an automated autostainer (Bond III, Leica Biosystems). Diaminobenzidine was used as a chromogenic substrate.

Estrogen and progesterone receptors were considered positive when at least 1% of tumor cell nuclei resulted immunoreactive. In accordance with the 2013 ASCO-CAP guidelines, Her2 overexpression was defined as a score of 3+ immunoreaction intensity or 2+ immunoreaction intensity with Her2 amplification by in situ hybridization [47].

The IHC expression of the protein biomarkers TAZ, YAP, AMPK, HMGCR, and SCD1 was evaluated in diagnostic core biopsies. For each biomarker, both the nuclear and cytoplasmic immunostainings were assessed by two investigators independently (C.E. and A.D.B.). Assessors were masked to treatment outcome. The raw ICH information was reported as nuclear (_N) and cytoplasmic (_C) staining for each biomarker expressed according to (i) the intensity of staining (_S), which was graded with a score of 0, 1, 2, or 3 for no color reaction, mild reaction, moderate reaction, or intense reaction, respectively, and (ii) the percentage of tumor cells (_P) displaying that immunoreaction, expressed in percentage from 0 (for 0% of the cells) to 100% (all the cells). We used these raw measures to calculate some composite variables according to the international immunoreactive score (SxP_irs) classification [48,49]. In more detail, the “percentage of cells stained” value was first categorized according to a 5-grade scale: 0 (no positive cells), 1 (<10% positive cells), 2 (10–50% positive cells), 3 (51–80% positive cells), and 4 (>80% positive cells). Then, it was multiplied by the immunoreaction intensity score (0, 1, 2, or 3) to produce the first composite variable, varying within a 0-to-12 range. The second composite variable was obtained by further categorizing this 13-grade scale variable into a 4-grade scale variable as follows: 0 = negative (0–1 SxP_irs score), 1 = positive weak (2–3 SxP_irs score), 2 = positive intermediate (4–8 SxP_irs score), and 3 = positive strong (9–12 SxP_irs score). The third composite variable was obtained by directly multiplying the percentage of cells stained and the immunoreaction intensity score, producing a new variable ranging from 0 to 300. We used all the raw and composite variables describing the immunostaining to assess their association with each other, with clinical–pathological features, and with pCR. The raw and composite variables along with the inherent labels are listed in Appendix B.

### 2.2. Statistical Analysis and Data Visualization

Descriptive statistics were used to summarize all the variables of interest. Continuous data are reported as mean or median estimates and ranges. Categorical features are represented with frequencies and percentage values. The associations between variables were assessed using Pearson’s Chi-squared test or Fisher’s exact test, as appropriate. The distribution of continuous variables with respect to categorical variables was compared by employing the Wilcoxon and Kruskal–Wallis tests. Correlations between clinical and immunohistochemical variables were explored by performing Pearson’s correlation test. In logistic regression models, we assessed the predictive potential of the different biomarkers and clinical features with respect to pCR. Variables that showed a statistically significant effect in univariate regressions were subsequently included in the multivariate logistic regression model. Statistical significance was set at *p*-values less than 0.05. For correlation tests, only those with |ρ| > 0.25 and *p*-value < 0.05 were considered.

Data visualization was accomplished by using violin plots, box plots, bar charts, and heatmaps.

Statistical analyses were carried out independently by two investigators (F.S. and E.K.) using SPSS software (SPSS version 21, SPSS Inc., Chicago, IL, USA) and R programming language (version 4.0.4). Data visualization was realized using the R programming language (version 4.0.4).

### 2.3. Machine Learning Algorithm

In the final phase of the analysis on the internal cohort, we also employed a shallow machine learning algorithm in order to try extricating among the biomarker IHC expression levels those with the highest relevance in predicting pCR. We used such an algorithm also with the intention to capture possible nonlinear effects/relationships between the biomarkers and pCR. We used scikit-learn (version 1.0.1) from the Python programming language (version 3.10.3) to fit a decision tree in our data that would predict pCR based on the biomarkers’ expressions, which were fed to the algorithm in the raw form. The “alpha” hyperparameter for decision tree pruning was optimized by iterating 5-fold cross-validation on 80% of the patients randomly selected for training and validation purposes. The remaining 20% of the data was used as a testing set to produce a confusion matrix. Finally, considering the limited size of our study sample, the definitive decision tree for pCR was fitted on the whole population using the previously optimized model.

### 2.4. TCGA Validation Using Survival Outcomes of Breast Cancer Patients

We selected the Breast Invasive Carcinoma—TCGA, PanCancer Atlas cohort for external validation purposes (https://www.cbioportal.org/ (accessed on 19 August 2022)). Among the protein expression data, we chose those obtained through reverse phase protein array (RPPA), available for a total of 876 breast cancer patients. Normalized RPPA data were used for the analysis. With respect to the proteins of interest for the present study, only measurements on TAZ, YAP, and SCD1 were available. Overall survival (OS) and disease-free survival (DFS) were used as clinical outcomes in the absence of data on neoadjuvant treatment and pCR. Additionally, data regarding hormonal receptor status were lacking. The pre-defined patient selection criteria were (i) Her2+ disease; (ii) stage I, II, or III; and (iii) available OS and DFS data. R programming language (version 4.0.4) was used for the survival analysis. The biomarkers of interest were tested singularly and as a common signature for impact on OS and DFS. Comparison was performed using the log-rank test, and statistical significance levels were set at 0.05. Results were visualized through survival curves built using the Kaplan–Meier method.

## 3. Results

### 3.1. Clinical Characteristics and Outcomes

Between August 2014 and March 2017, 65 Her2+ positive BC patients were included in the study. The demographics, anthropometrics, and key clinic-pathological features of the study population are listed in Table 1.

The median age at diagnosis was 49 years (range: 34–78). Twenty-seven (41.5%) patients were premenopausal, and 38 (58.5%) were postmenopausal. Based on the cut-off points established by the Centers for Disease Prevention (https://www.cdc.gov/obesity/basics/adult-defining.html, last accessed on 16 June 2022) [50], 21 (32.3%) patients had a normal body mass index (BMI), 3 (4.6%) were in the underweight range, 16 (24.6%) were overweight, 7 (10.8%) were obese, and BMI was not reported for 18 (27.7%) of them.

The most common disease stage at diagnosis was stage II (60.0%), followed by stage III (33.8%) and stage I (4.6%). Overall, 59 (90.7%) patients had ER-positive (ER+) tumors, while two (3.1%) patients had ER-negative (ER-) tumors; in four (6.2%) patients, the ER status resulted unknown. Regarding the PR status, breast tumors were positive in 44 (90.7%) patients, negative in 17 (26.2%), and unknown in 4 (6.2%) cases. Only two (3.1%) patients had ER- and PR-negative tumors, while in 59 (90.7%) patients, tumors were positive for at least one hormonal receptor. Overall, the Her2 IHC score was 3+ in 46 (70.8%) patients, 2+ in 15 (23.1%) patients, and not reported in 4 (6.2%) patients. All cases with a Her2 IHC score of 2+ or unknown showed amplified Her2 in the in situ hybridization (ISH) testing. The expression of Ki67 was high (defined as >20%) in 49 (75.4%) patients, low (defined as ≤20%) in 13 (20%) patients, and unknown in the remaining 3 (4.6%) patients. For all the patients included, trastuzumab, anthracyclines, and taxanes were part of the treatment regimen. The median number of neoadjuvant cycles administered was eight (range, 6–8). In specific regard to the study endpoint, a total of 19 (29%) patients obtained a pCR after neoadjuvant treatment and subsequent surgery, whereas 46 (71%) patients showed residual disease at pathologic assessment after definitive surgery (Figure 2A).

### 3.2. Expression and Associations between Target Protein Biomarkers (TAZ, YAP, AMPK, HMGCR, and SCD1)

In terms of the percentage of cells stained, all the biomarkers tested except AMPK and TAZ showed a clear prevalence of cytoplasmic expression (Table 2A). The same pattern was also observed in terms of immunoreaction intensity score (Table 2B). The use of the composite variable obtained by multiplying the percentage with the intensity score confirmed that YAP, HMGCR, and SCD1 were more expressed in the cytoplasm. Conversely, AMPK was more expressed in the nucleus, while TAZ tended to be more equally expressed (Table 2C). These same expression patterns are graphically displayed in Figure 2B–D.

Correlations between the biomarkers’ expression tested either as single or composite variables are shown in Figure 3A, which also encodes the correlation coefficients and statistical significance levels. The exact Pearson’s correlation coefficients (ρ) and relative *p*-values are listed in Appendix A, respectively.

Overall, there was mostly a tendency of positive correlation between the biomarker expressions. This positive correlation was statistically significant between cytoplasmic YAP expression and cytoplasmic (*p* < 0.01) and nuclear (*p* < 0.05) TAZ expression, and between cytoplasmic TAZ expression and nuclear SCD1 expression (*p* < 0.05). Conversely, statistically significant negative correlations were observed between cytoplasmic TAZ expression and nuclear YAP expression (*p* < 0.05).

### 3.3. Associations between Clinical–Pathological Features and Target Protein Biomarkers (TAZ, YAP, AMPK, HMGCR, and SCD1)

Positive correlations were observed between ER and PR expression (*p* < 0.001) and between Ki67 expression and histologic grade at diagnosis (G) (*p* < 0.05). Ki67 % expression was also positively correlated with nuclear YAP expression (*p* < 0.05). Stage at diagnosis was positively correlated with BMI (*p* < 0.01). On the other hand, BMI was negatively correlated with nuclear AMPK expression (*p* < 0.05) and nuclear YAP expression (*p* < 0.05). Lastly, there was also a negative correlation between ER expression and nuclear HMGCR expression (*p* < 0.05). All the mentioned correlations are represented in Figure 3A.

Further significant associations with potential biological implications were observed using separate statistical comparisons. In tumors of patients placed within the normal and overweight BMI categories, cytoplasmic YAP expression was more marked compared to patients in the underweight and obese categories (Kruskal–Wallis *p* < 0.01). ER-negative cancers exhibited higher cytoplasmic TAZ (Kruskal–Wallis *p* = 0.04). Lower Ki67 was associated with higher expression of cytoplasmic SCD1 (Kruskal–Wallis *p* = 0.04). Stage I cancers showed higher expression of nuclear HMGCR (Kruskal–Wallis *p* = 0.015) and cytoplasmic TAZ (Kruskal–Wallis, *p* = 0.04). Stage III tumors showed higher expression of cytoplasmic SCD1 (Kruskal–Wallis, *p* < 0.01).

### 3.4. The Impact of Biomarker Expression on pCR

We first compared the distribution of biomarker expression between the tumors of patients that reached pCR and those who did not. Overall, no clear-cut differences emerged in terms of immunoreaction intensity score in patients who reached pCR compared to patients with residual disease (Figure 3B). Statistical comparison by the Wilcoxon test of the expressions in terms of the composite variable SxP showed a significant difference between the two groups only in the case of nuclear HMGCR expression. In fact, nuclear HMGCR expression appeared significantly higher in patients who obtained pCR when compared to the patients that did not (*p* = 0.03) (Figure 3C).

Subsequently, clinical–pathological variables including age, menopausal status, BMI, stage, grading, ER levels, PR levels, Her2 IHC score, Ki67, and treatment schedule (Table 1) and molecular variables including all the expression levels of TAZ, YAP, AMPK, HMGCR, and SCD1, expressed in terms of raw values and composite values, were tested as predictors of pCR in the logistic regression. The variables that tested significant in univariate models for their effect on pCR status are shown in Figure 4A.

Estrogen receptor expression > 70% and BMI > 20 were associated with significantly lower chances of pCR (OR 0.18 (95% CI: 0.05–0.59), *p* < 0.006, and OR 0.15 (95% CI: 0.02–0.82), *p* = 0.038, respectively). In addition, tumors showing higher expression of cytoplasmic TAZ based on the _SxP_irs_c classifier had more than five times higher odds of reaching pCR (OR 5.12 (95% CI: 1.12–27.65), *p* = 0.039). In terms of pCR rate, its magnitude was 24.6% in low TAZ tumors versus 62.5% in high TAZ tumors. Differences in the distribution of cytoplasmic TAZ in terms of *_*SxP_irs_c classes by pCR outcome are also graphically displayed in Figure 4B. However, when included in a multivariate model, the only variable which retained its statistical significance was ER > 70% (OR 0.14 (95% CI: 0.03–0.55), *p* < 0.005) (Figure 4C).

Figure 5 displays representative images of the immunostaining for TAZ as the main predictor of pCR. Expression levels were quantified according to the international immunoreactive score categories. The tumors in Figure 5A,B are from patients that did not obtain pCR, while the tumors in Figure 5C,D are from patients that reached pCR. The expression level of TAZ in Figure 5A is “negative” in both the nucleus and cytoplasm. The TAZ expression levels in Figure 5B are “positive weak” in both the nucleus and cytoplasm. Conversely, the TAZ expression level in Figure 5C is “positive intermediate” in both the nucleus and cytoplasm, while in Figure 5D, the expression level is “positive strong” in the nucleus and “positive intermediate” in the cytoplasm.

### 3.5. Machine Learning-Based Discovery of Biomarker Signatures Predictive of pCR

With the purpose of seeking a biomarker signature and considering the possible collinearities between independent variables with consequent nonlinear effects on pCR, we used a machine learning algorithm for a more accurate analysis. This was also useful considering the fact that we had a limited study sample relative to the number and heterogeneity of features that were analyzed, and the risk of overfitting was high. We chose decision trees, which we considered an appropriate shallow learning algorithm for this case with acceptable interpretability and low probability of overfitting after careful fine-tuning of the “alpha” parameter based on five-fold cross-validation of the model. As input predictor variables, we plugged in all the biomarker expressions as raw values. The final decision tree that we obtained is shown in Figure 6.

The main nodes of the tree show that biomarkers’ expression in terms of intensity of staining score can predict the absence, rather than the presence, of pCR—i.e., it is more reliably associated with residual disease rather than pCR accomplishment. More specifically, an HMGCR immunoreaction intensity score of 0 in the nucleus; a YAP immunoreaction intensity score of 0 or 1 in the nucleus; an SCD1 immunoreaction intensity score of 1, 2, or 3 in the cytoplasm; and a TAZ immunoreaction intensity score of 0 or 1 in the cytoplasm were all nodes with high discerning potential towards residual disease following neoadjuvant treatment at definite surgery.

These four markers were included into a signature, which was tested for its predictive role for pCR. Thirty-two patients had tumors testing positive for this signature, which, in the univariate analysis, was associated with a significantly reduced chance of achieving pCR (OR 0.11 (95% CI: 0.02–0.39), *p* = 0.002). The pCR rate in patients whose tumors carried this signature was 9.4% versus 51.6% in patients testing negative for the same signature. When tested in multivariate models also including ER > 70% and BMI > 20, the signature was no longer statistically significant (Figure 7A,B).

To explore collinearity between the signature and the other two variables, we tested the presence of the signature in conditions differing by ER and BMI. Indeed, the signature was more expressed in patients characterized by ER > 70% and BMI > 20 (in Figure 7C,D). Both of these features are less common in patients achieving pCR, as well as in patients whose disease most commonly expresses the signature. However, the distribution of the signature across categories defined by the preset cut-off values of ER and BMI was not statistically significant.

### 3.6. Testing Our Signature in the TCGA Cohort for Validation in Terms of Survival Outcomes

With the intent of validating our findings about the signature externally, we used the clinical and proteomic data of the Breast Invasive Carcinoma—TCGA, PanCancer Atlas dataset. The entire cohort comprised 1084 breast cancer patients, and protein expression profiling data of the primitive tumors were available in 876 cases. We chose RPPA expression levels and used the normalized values. With respect to the proteins constituting our signature, the TCGA dataset provided information only on TAZ, YAP, and SCD1. Similarly, the dataset did not provide information related to hormonal receptor status and neoadjuvant treatment. After applying selection criteria for patients with localized/locally advanced disease, we identified a total of 56 Her2+ positive cases. Data on OS were available for all 56 patients, while DFS was provided for only 49 of them. Using R programming, we applied an automatized method to identify the best cut-off value for TAZ, YAP, and SCD1 for classifying patients as long survivors or short survivors. The best cut-offs for each biomarker resulted as follows: TAZ, 43rd percentile; YAP, 27th percentile; and SCD1, 40th percentile. We then categorized the expression levels as low if they were lower than or equal to these cut-offs or high if they were higher. Subsequently, we built a signature resembling the one we used from our cohort; that is, if TAZ or YAP was low and SCD1 was high, we considered the tumor as having the signature—i.e., the signature was “present”. Otherwise, the signature was “absent”. We then tested this signature for its impact on OS and DFS. Patients that had tumors with the signature “present” had a significantly lower OS when compared to those with the signature “absent” (Figure 8A). In particular, the 72-month survival rate was 37% when the signature was “present” and 90% when it was “absent” (*p* = 0.02). We also tested the same signature with DFS, but in this case, no relevant differences were detected between patients with tumors with the signature “present” and those with the signature “absent” (*p* = 0.78), as shown in Figure 8B.

## 4. Discussion

In this prospective observational study, key actors of the Hippo pathway and central players of cell metabolism were tested in reference to pCR rate in 65 Her2+ BC patients treated with trastuzumab-based neoadjuvant therapy.

The results showed that lower IHC expression of TAZ in tumor cells was associated with a lower probability of obtaining pCR, especially if this condition was concomitant with lower expression of YAP and HMGCR and higher expression of SCD1. In more detail, the presence in the tumors of a signature including a cytoplasmic TAZ immunoreaction intensity score ≤ 1, a nuclear YAP immunoreaction intensity score ≤ 1, a nuclear HMGCR immunoreaction intensity score = 0, and a cytoplasmic SCD1 immunoreaction intensity score > 0 was highly predictive for residual disease following neoadjuvant therapy and surgery.

Unfortunately, when included in multivariate models along with variables testing significant in univariate analyses, i.e., ER > 70% and BMI > 20, this signature was no longer predictive of pCR.

However, when testing a similar protein signature to ours in 56 Her2+ patients from the TCGA cohort, we found a concordant outcome with respect to our study, but in terms of OS. Specifically, we observed that patients who had tumors with low TAZ or YAP and high SCD1 had a poor prognosis in terms of OS compared to patient with tumors that did not show the same expression profile of the biomarkers. Indeed, a lower pCR rate is a proxy of a worse prognosis, which could make our findings concordant with those of the TCGA in this sense. However, the same signature did not predict DFS in the Her2+ TCGA cohort, probably because the number of patients tested was even lower (N = 49).

We previously tested the Hippo pathway in reference to pCR rate in Her2+ BC patients from a neoadjuvant setting [46]. Within the retrospective study including data on 61 patients, we were unable to identify biomarkers predictive of pCR for the overall study population. Conversely, following stratification by Her2−enriched vs. luminal B (Her2+) BC patients, in 35 patients in the luminal B subgroup, significantly higher pCR rates were observed in patients with lower TAZ expression (82% vs. 44% pCR rates, *p* = 0.035). The results remained significant when exclusively restricting the set of analysis to the triple positive BC patients, i.e., to patients with a Her2+ disease and with both ER and PR percent cell expression ≥ 50% (*p* = 0.035). The results of the previous retrospective study are not concordant with the current study’s findings. Although fully comparable by study population size, i.e., 65 vs. 61 patients enrolled, these two studies sensibly differ in many respects. In the retrospective study, luminal B tumors represented 57.4% of the overall study sample, whereas the percentage of patients with hormone receptor-positive tumors was 91% in the current study. Of further relevance, in our previous study, BC patients with stage III cancer at diagnosis represented more than 61% of the cases, as opposed to 33% in the present prospective study. It is plausible that differences in these key clinical–pathological features may at least partially explain differences in terms of pCR rate, i.e., 67% vs. 29%; predictive values of TAZ; and overall study results across these two studies.

Indeed, previous studies have suggested that ER-positivity is a negative predictive factor of pCR and that the crosstalk between the ER and Her2 pathways is involved in the onset of drug resistance [51]. Furthermore, pivotal neoadjuvant randomized clinical trials such as NeoALTTO [52], GEPARQUINTO [53], and NeoSphere [54] confirmed that positivity for hormone receptors reduced the likelihood of achieving pCR.

In the present study, besides the canonical transducers TAZ and YAP, we tested the hypothesis of a predictive effect of AMPK, HMGCR, and SCD1 for pCR. To our knowledge, this is the first prospective study to jointly investigate these biomarkers related to the Hippo pathway and cell metabolism regulation in the same patient cohort. However, we found other reports in which the prognostic and/or predictive effect of these biomarkers was assessed separately in different experimental settings. In a previous study including 14 Her2+ BC patients treated with trastuzumab-based neoadjuvant therapy, YAP-positive tumors showed a 57.1% pCR rate as compared to 0% in their YAP-negative counterparts [55]. This finding is consistent with our results with respect to the low YAP expression within the signature having negative predictivity for pCR. Conversely, in another study on 397 Her2+ BC patients, higher values of a common TAZ/YAP score were predictive of a lower pCR rate (17% and 31% in high and low score cases, respectively) [56]. We did not identify any previous study assessing the role of HMGCR and SCD1 in the neoadjuvant setting of Her-2+ BC. However, one study found some evidence of low HMGCR-positive IHC expression to be associated with worse prognosis in Her2 score 2+ and 3+ BC patients [57]. The inclusion of Her2 2+ BC patients reduced the comparability between results from the two studies.

The association of TAZ/YAP expression with treatment outcomes is thus far controversial. Indeed, nuclear activity of TAZ and YAP as transcriptional regulators induces the activation of target genes which favor S-phase entry and mitosis in BC cells [58]. Furthermore, TAZ/YAP activity favors cancer cell stemness, which is an accepted factor of chemoresistance [59]. On this basis, higher TAZ/YAP expression would be more consistent with a lower response to treatment. Nevertheless, several studies showed that TAZ/YAP activation could also translate into tumor-suppressing effects depending on the context and on the specific signal rewiring that was established during tumor development. For instance, TAZ overexpression in hematological malignancies translates into a tumor-suppressing factor by inhibiting MYC [60]. Other studies have demonstrated that both TAZ and YAP expression in solid tumors promotes ferroptosis, which is an emerging tumor suppressor mechanism [61,62]. Moreover, TAZ and YAP are coactivator proteins that lack DNA-binding activity and need to interact with transcription factors to influence gene expression. In this sense, even though the main DNA-binding partners of TAZ/YAP are the TEAD family proteins, which activate genes involved in cell proliferation, interaction with other transcription factor partners having tumor suppressor effects is possible and has been described. For example, a study showed that YAP can interact and coactivate DNA-binding tumor suppressors such as RUNXs and p73 (which respectively mediate cell differentiation and apoptosis), therefore acquiring an inhibitory role towards tumor growth [63]. One of the studies that was mentioned above on Her2+ BC patients confirmed that YAP increases the response to trastuzumab-based neoadjuvant therapy by facilitating p73-induced apoptosis [55]. Taken together, these studies invite further investigations with a quite complex design including both pre-clinical and clinical tasks.

An additional aspect in need of critical discussion is that neither TAZ expression alone nor the TAZ/YAP/HMGCR/SCD1 signature maintained statistical significance when tested together with ER (>70% vs. Others) and BMI (>20 vs. Others). This may be at least partly due to the relatively limited sample size, which may reduce the statistical power for the associations tested. Given the full availability of two databases (DBs) from the studies coordinated by our center, we are currently planning an individual patient data meta-analysis which may significantly extend our statistical power. In addition, survival data for both studies are in a course of update, which preludes a first set of survival analyses, considered, thus, inadequate for the potentially immature outcomes.

Interestingly, besides ER and BMI, we found no predictive effect for pCR of other clinical–pathological features, such as age, stage, grade, PR, Ki67, menopausal status, and treatment regimens used. However, our small sample size and the unbalanced distribution of these characteristics in our cohort make these results inconclusive.

In this study, we dedicated special efforts to distinguish the cytoplasmic versus nuclear localization of the biomarkers’ expression to fully exploit the advantage of using immunohistochemistry as compared to other methods of protein quantification. Overall, nuclear TAZ/YAP localization corresponds to a condition in which the two transducers are exerting their effects, while cytoplasmic localization could imply their inactivation by proteasomal degradation or, alternatively, their permanence in a storage site before being shuttled into the nucleus [19,20,21,22,23,24,25]. In our experiments, the cytoplasmic expression was of a higher magnitude as compared to the nuclear expression for the majority of the biomarkers, but not for TAZ, which showed similar expression in the two compartments, and AMPK, which was more expressed in the nucleus. The prevalent nuclear localization for AMPK could be explained by its role as a p53-dependent metabolic checkpoint of the cell cycle, which occurs in the nucleus [64]. In this context, it is relevant that in our negative predictive signature, the core component scored low in cytoplasmic TAZ expression, while the additional conditions that increased the discriminating potential were a low nuclear score of YAP/HMCGR and a high cytoplasmic score of SCD1. Therefore, dedicated studies might be necessary for a more accurate clarification of the role of the cytoplasmic and nuclear expression of these biomarkers.

Some aspects of our study design might translate into limitations. We have previously mentioned the limited sample size and current lack of evidence from survival data, along with our orientation toward the use of more refined statistical methods to allow quantitative data synthesis and a prompt update of survival data for the performance of a survival analysis.

Our study also has several strengths. First, data were prospectively collected. The IHC measurements of the TAZ, YAP, HMGCR, AMPK, and SCD1 protein biomarkers were centralized at our coordinator center and performed independently by two experts, who were blinded to each other’s results, the patients’ clinical–pathological features, and the study outcomes. Moreover, the statistical analysis was performed independently by two authors, using different computer tools. Lastly, we applied to our data analysis the most advanced data science techniques, consisting of machine learning algorithms.

Overall, our study provides additional cues on the elements that can help build a therapeutic strategy informed by molecular biomarkers to yield to an increasingly tailored treatment plan at the single patient level.

Within a quite productive research pipeline, which still flourishes thanks to our network of collaborating cancer centers operating at the national level, we intend to further ameliorate aspects related to the limitations in the current statistical power and use more refined methods to perform a quantitative synthesis of the data from our two dedicated DBs. Survival endpoints will shortly be considered for further testing of the predictive potentials of the biomarkers of interest.

## 5. Conclusions

Overall, we prospectively enrolled 65 patients affected by localized/locally advanced Her2+ BC and treated with trastuzumab-based neoadjuvant therapy followed by surgery in order to evaluate the predictive value of Hippo pathway transducers (TAZ/YAP) and specific cell metabolism regulators (AMPK/HMGCR/SCD1) for pCR. The results did not suggest a clear impact of the tumor IHC expression of these biomarkers on the main outcome of interest. First, the negative predictive potential with respect to pCR of low cytoplasmic levels of TAZ, although strengthened by concomitant low nuclear expression of YAP/HMGCR and high cytoplasmic expression of SCD1, was not significant in the multivariate analysis. Second, our current results are controversial when compared to our previous findings and work from other research groups. In interpreting our results, the relatively low comparability of the current study with prior studies from our group and other groups deserves consideration. Similarly, its relative strengths, mainly exemplified by its prospective design and use of advanced analytic methods, should not be neglected. Additionally, we obtained concordant results with the external TCGA dataset. In future investigations, the enrolment of an adequately sized patient population and the accomplishment of both clinical and pre-clinical tasks contemplating wider omics profiling at the transcriptomic and/or genomic level are warranted.

## Figures and Tables

**Figure 1 cancers-14-04835-f001:**
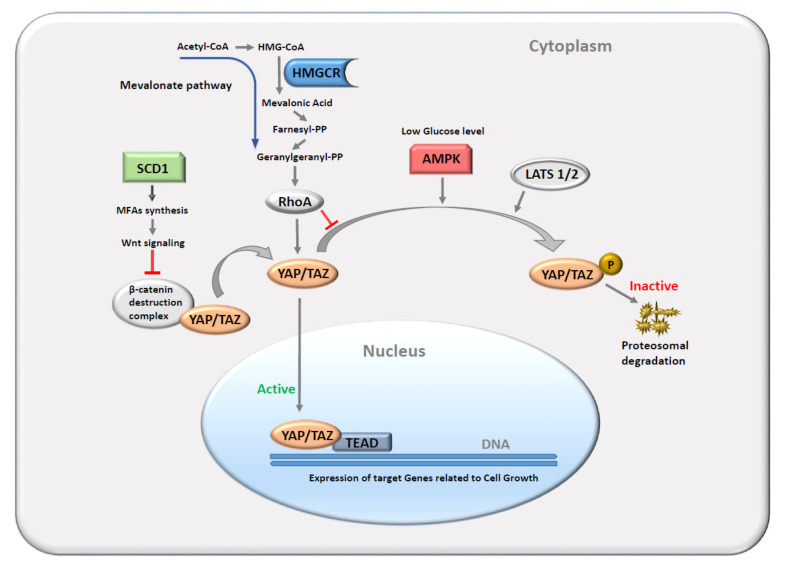
A simplified scheme of the main Hippo pathway kinases and effectors and the modulating effects from main cellular metabolism regulators. Only core components of interest in our study are shown in colored shapes. The effectors YAP and TAZ are inactivated after LATS1/2-facilitated phosphorylation or, alternatively, enter the nucleus and bind to TEAD transcription factors activating the expression of genes related to cell growth. Low levels of glucose, acetyl-CoA, and MFAs favor YAP/TAZ inactivation through the AMPK, HMGCR-RhoA, and SCD1-Wnt-β-catenin pathways, respectively. Abbreviations: YAP, Yes-associated protein; TAZ, transcriptional coactivator with PDZ-binding motif; AMPK, AMP-activated protein kinase; HMGCR, HMG-CoA reductase; RhoA, Ras Homolog Family Member A; SCD1, Stearoyl-CoA-desaturase 1; MFAs, monounsaturated fatty acids.

**Figure 2 cancers-14-04835-f002:**
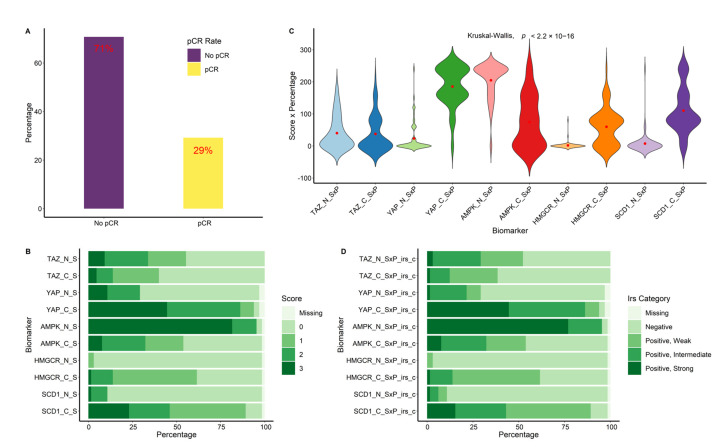
Panel (**A**) shows the rates of pathological complete response (pCR) and No pCR in the whole study population. The expression levels of the biomarkers in the nucleus (_N) and in the cytoplasm (_C) are shown in Panel (**B**) in terms of the immunoreaction intensity score (_S), in Panel (**C**) in terms of multiplication of the immunoreaction intensity score with the percentage of cells stained (_SxP), and in Panel (**D**) in terms of the categorized international immunoreactive score (SxP_irs_c).

**Figure 3 cancers-14-04835-f003:**
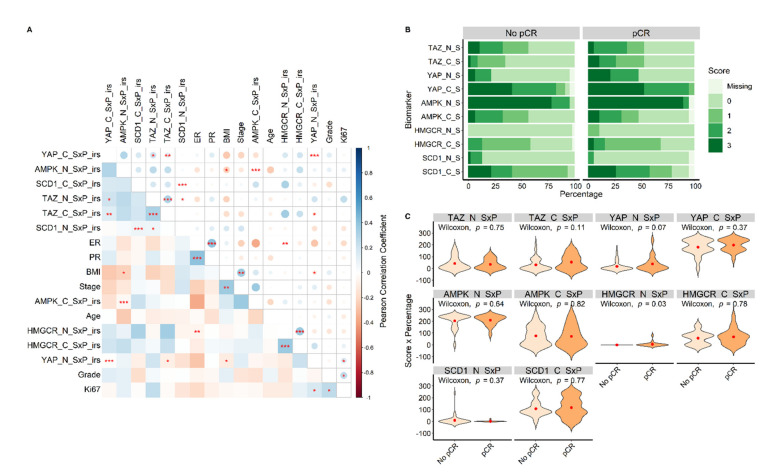
The correlation heatmap showing comparisons between all biomarkers and relevant clinical features is displayed in Panel (**A**). Blue color represents positive correlation coefficients, while red color represents negative correlation coefficients. The intensity of the color in the whole panel and the dimension of the circles in the upper triangle represent the magnitude of the correlation coefficients. Statistical significance levels are shown by asterisks as follows: * for *p* < 0.05, ** for *p* < 0.01, and *** for *p* < 0.001. The expression proportions of biomarkers in tumors that reached pCR compared to tumors that did not reach pCR (No pCR) are displayed in terms of percentage of cells stained (_P) in Panel (**B**) and in terms of immunoreaction intensity score (_S) in Panel (**C**). Suffixes: _N = nuclear; _C = cytoplasmic; _S = immunoreaction intensity score; _P = percentage of cells stained; _SxP = multiplication of immunoreaction intensity score with percentage of cells stained; SxP_irs = international immunoreactive score.

**Figure 4 cancers-14-04835-f004:**
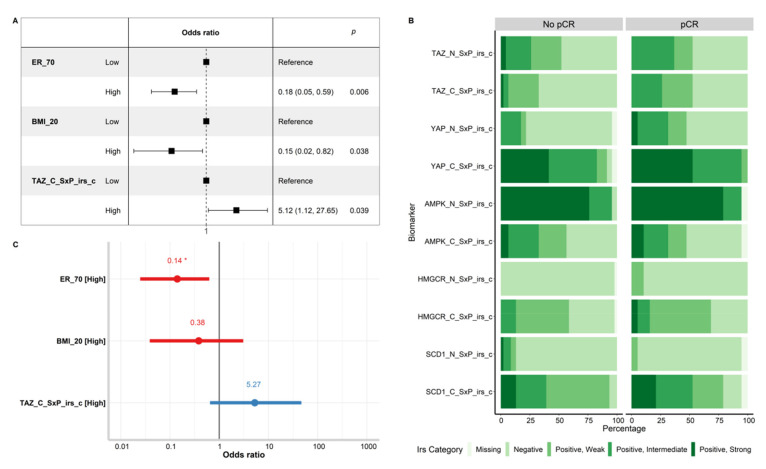
Panel (**A**) displays the univariate logistic model for the variables that showed a statistically significant effect on pCR. Panel (**B**) shows the stacked bar plots of the biomarker expression level in terms of categorized international immunoreactive score (SxP_irs_c), comparatively for tumors that reached pCR and those which did not (No pCR). Panel (**C**) shows the multivariate logistic model which includes all the variables showing a significant impact on pCR in the univariate model. Abbreviations: ER_70 = estrogen receptor with cut-off at 70% (ER > 70% vs. Others); BMI_20 = body mass index with cut-off of 20 (BMI > 20 vs. Others). Suffixes: _C = cytoplasmic. In Panel (**C**), * represents *p* < 0.05, as the only statistical significance level.

**Figure 5 cancers-14-04835-f005:**
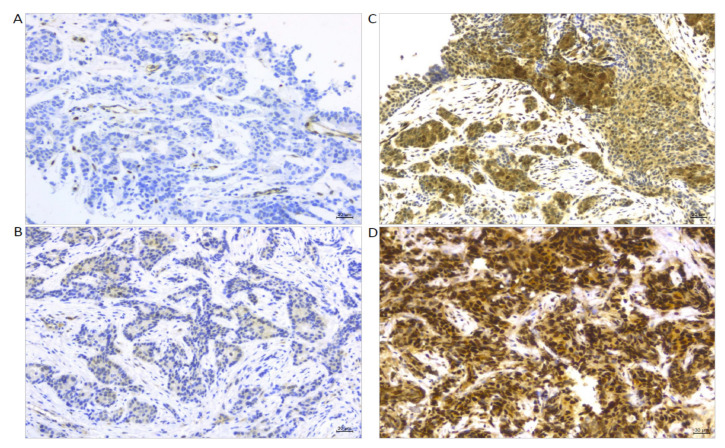
TAZ expression levels in terms of international immunoreactive score categories (SxP_irs_c). (**A**) Tumor from a patient that did not obtain pCR, showing “negative” nuclear and cytoplasmic expression. (**B**) Tumor from a patient that did not obtain pCR, showing “positive weak” nuclear and cytoplasmic expression. (**C**) Tumor from a patient that obtained pCR, showing “positive intermediate” nuclear and cytoplasmic expression. (**D**) Tumor from a patient that obtained pCR, showing “positive strong” nuclear and “positive intermediate” cytoplasmic expression.

**Figure 6 cancers-14-04835-f006:**
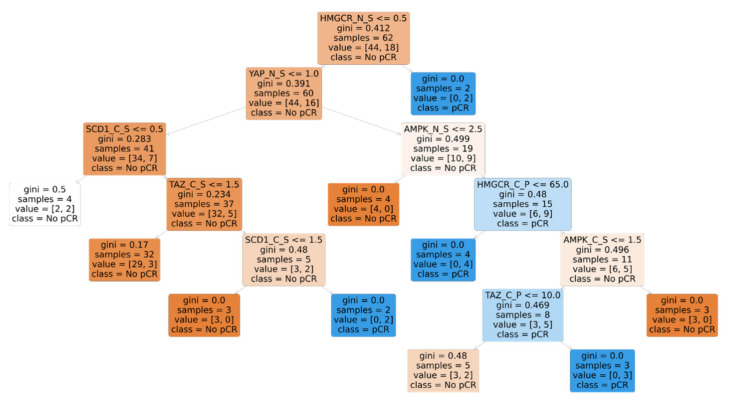
Decision tree that fits with hyperparameter alpha = 0.03 and with all the raw biomarker expressions as predictors of pCR. Tree interpretation: right branching = No; left branching = Yes; gini = node or leaf purity. Suffixes: _N = nuclear; _C = cytoplasmic; _S = immunoreaction intensity score; _P = percentage of cells stained.

**Figure 7 cancers-14-04835-f007:**
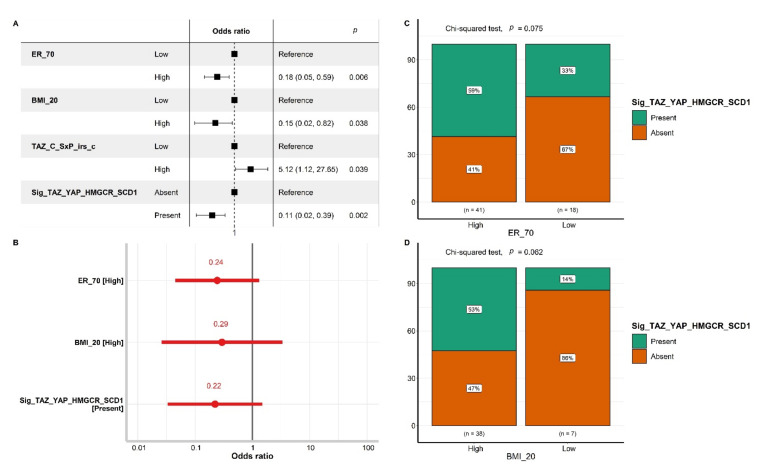
(**A**) The univariate logistic model for the variables that showed a statistically significant effect on pCR. (**B**) The multivariate logistic model which includes all the variables showing a significant impact on pCR in the univariate model. (**C**,**D**) The distribution of Sig_TAZ_YAP_HMGCR_SCD1 in the two categories of ER (ER > 70% vs. Others) and BMI (BMI > 20 vs. Others), respectively. Statistical comparison was performed using the Chi-squared test. Abbreviations: ER_70 = estrogen receptor with cut-off at 70% (ER > 70% vs. Others); BMI_20 = body mass index with cut-off of 20 (BMI > 20 vs. Others); Sig_TAZ_YAP_HMGCR_SCD1 = signature of cytoplasmic TAZ score 0–1, nuclear YAP score 0–1, nuclear HMGCR score 0, and cytoplasmic SCD1 score 1–3. Suffixes: _C = cytoplasmic; SxP_irs_c = categorized international immunoreactive score.

**Figure 8 cancers-14-04835-f008:**
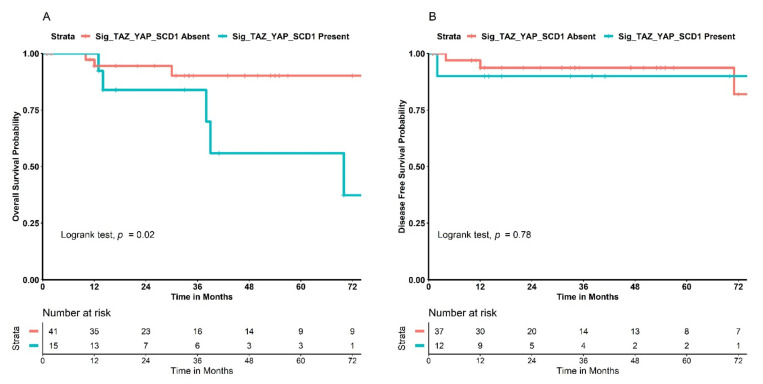
Validation of a similar signature to the one we identified on breast cancer patients from the Breast Invasive Carcinoma—TCGA, PanCancer dataset. (**A**) Kaplan–Meier curves and risk table displaying the comparison in terms of overall survival between patients with tumors with the signature “present” and those with tumors with the signature “absent”. (**B**) Kaplan–Meier curves and risk table displaying the comparison in terms of disease-free survival between patients with tumors with the signature “present” and those with tumors with the signature “absent”. Abbreviations: Sig_TAZ_YAP_SCD1: signature of reverse phase protein array low TAZ or YAP levels and high SCD1 levels.

**Table 1 cancers-14-04835-t001:** Characteristics of the study participants (N = 65).

Characteristics	N (%)
	65(100)
**Age in years** (median, range)	49 (34–78)
**Menopausal status** Premenopausal Postmenopausal	27 (41.5) 38 (58.5)
**^1^BMI** Underweight (<18.5) Normal (18.5–25) Overweight (25.1–30) Obese (>30.1) Missing	3 (4.6) 21 (32.3) 16 (24.6) 7 (10.8) 18 (27.7)
**Stage** I II III Missing	3 (4.6) 39 (60.0) 22 (33.8) 1 (1.5)
**Grading** G1 G2 G3 Missing	2 (3.1) 25 (38.5) 30 (46.2) 8 (12.3)
**Estrogen Receptor** Positive Negative Missing	59 (90.7) 2 (3.1) 4 (6.2)
**Progesterone Receptor** Positive Negative Missing	44 (90.7) 17 (26.2) 4 (6.2)
**Her2 IHC score** Score 2+ Score 3+ Missing	15 (23.1) 46 (70.8) 4 (6.2)
**Ki67 continuous** (median, range)	30 (1–80)
**Ki67** >20 ≤20 Missing	49 (75.4) 13 (20) 3 (4.6)
**Neoadjuvant Treatment Schedule** AC → P (or D) + T AC + T → P (or D) + T P (or D) + T → AC + T	47 (72.3) 14 (21.5) 4 (6.2)

Abbreviations: N, number; BMI, body mass index; IHC, immunohistochemistry; AC, anthracycline cyclophosphamide; P, paclitaxel; D, docetaxel; T, trastuzumab. ^1^BMI: body mass index. BMI categories are defined according to the cut-off points set by the Centers for Disease Prevention, as available at https://www.cdc.gov/obesity/basics/adult-defining.html, last accessed on 16 June 2022.

**Table 2 cancers-14-04835-t002:** Cytoplasmic and nuclear expression of TAZ, YAP, AMPK, HMGCR, and SCD1 (N = 65), according to (**A**) mean percentage of cells stained, (**B**) intensity of staining on a 0-3 scale, and (**C**) immunoreactive score classification.

A					
**Biomarker**	**^1^ P** **(% of Cells)**	**^1^ SxP *** **(% of Cells * Intensity (0–3))**		**^1^ Irs** **% of Cells (in 0–4 Scale) * Intensity (0–3))**	
TAZ_C TAZ_N	27 22	38 40		1.6 2.1	
YAP_C YAP_N	76 9	185 23		7.0 1.3	
AMPK_C AMPK_N	43 71	75 205		2.8 8.1	
HMGCR_C HMGCR_N	48 2	60 2		2.3 0.1	
SCD1_C SCD1_N	65 3	110 8		4.4 0.4	
**B**					
**Biomarker**	**0–3 Score**				
	**0** **Nr (%)**	**1** **Nr (%)**	**2** **Nr (%)**	**3** **Nr (%)**	**Missing** **Nr (%)**
TAZ_C TAZ_N	39 (60) 29 (44.6)	17 (26.2) 14 (21.5)	6 (9.2) 16 (24.6)	3 (4.6) 6 (9.2)	0 (0) 0 (0)
YAP_C YAP_N	2 (3.1) 44 (67.7)	5 (7.7) 0 (0)	27 (41.5) 12 (18.5)	29 (44.6) 7 (10.8)	2 (3.1) 2 (3.1)
AMPK_C AMPK_N	29 (44.6) 2 (3.1)	14 (21.5) 0 (0)	16 (24.6) 9 (13.8)	5 (7.7) 53 (81.5)	1 (1.5) 1 (1.5)
HMGCR_C HMGCR_N	24 (36.9) 62 (95.4)	31 (47.7) 2 (3.1)	8 (12.3) 0 (0)	1 (1.5) 0 (0)	1 (1.5) 1 (1.5)
SCD1_C SCD1_N	6 (9.2) 57 (87.7)	28 (43.1) 0 (0)	15 (23.1) 6 (9.2)	15 (23.1) 1 (1.5)	1 (1.5) 1 (1.5)
**C**					
**Biomarker**	**Irs Classification**				
	**Negative** **Nr (%)**	**Positive,** **Weak** **Nr (%)**	**Positive, Intermediate** **Nr (%)**	**Positive,** **Strong** **Nr (%)**	**Missing** **Nr (%)**
TAZ_C TAZ_N	40 (61.5) 31 (47.7)	17 (26.2) 15 (23.1)	7 (10.8) 17 (26.2)	1 (1.5) 2 (3.1)	0 (0) 0 (0)
YAP_C YAP_N	2 (3.1) 44 (67.7)	5 (7.7) 5 (7.7)	27 (41.5) 13 (20)	29 (44.6) 1 (1.5)	2 (3.1) 2 (3.1)
AMPK_C AMPK_N	29 (44.6) 2 (3.1)	14 (21.5) 0 (0)	16 (24.6) 12 (18.5)	5 (7.7) 50 (76.9)	1 (1.5) 1 (1.5)
HMGCR_C HMGCR_N	24 (36.9) 62 (95.4)	31 (47.7) 2 (3.1)	8 (12.3) 0 (0)	1 (1.5) 0 (0)	1 (1.5) 1 (1.5)
SCD1_C SCD1_N	6 (9.2) 57 (87.7)	30 (46.2) 3 (4.6)	18 (27.7) 3 (4.6)	10 (15.4) 1 (1.5)	1 (1.5) 1 (1.5)

^1^ Reported percent values are means. * SxP is a composite variable obtained by multiplying the percentage of cells stained and the immunoreaction intensity score. Abbreviations: P, percentage; S, score; Irs, immunoreactive score; TAZ, transcriptional coactivator with PDZ-binding motif; YAP, Yes-associated protein; AMPK, AMP-activated protein kinase; HMGCR, HMG-CoA reductase; SCD1, Stearoyl-CoA-desaturase 1. Suffixes: _C = cytoplasmic; _N = nuclear.

## Data Availability

The datasets generated and analyzed during the current study are available in the GARRbox repository (https://gbox.garr.it/garrbox/index.php/s/akodeBvq5ihxoAa (accessed on 31 August 2022)).

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
