# Peer review of "The Impact of the Hippo Pathway and Cell Metabolism on Pathological Complete Response in Locally Advanced Her2+ Breast Cancer: The TRISKELE Multicenter Prospective Study"

_cancers, 2022, doi:10.3390/cancers14194835_

Round 1

Reviewer 1 Report (Previous Reviewer 1)

Following the revision, it is much more better and impactful MS. However minor revision is needed before acceptance.

There are certain inconsistency in  figures (especially in the font size). In figure 4 and figure 7, font sizes are not matching with rest of the figures. 

Authors may shorter the discussion.

This manuscript is a resubmission of an earlier submission. The following is a list of the peer review reports and author responses from that submission.

Round 1

Reviewer 1 Report

The importance of YAP/TAZ signaling in HER2+/BC treatment outcome is very controversial and extremely complex due to their expression pattern i.e. cytoplasmic versus nuclear expression. This is a prospective study (although number of patients is not high) and this is the only plus point of the study.

Critical comments:

1. Very difficult to flow because of the writing style. Authors may change the writing pattern like paragraph by paragraph according to the signaling protein. Better not to mix up the YAP/TAZ signaling with metabolic protein.

2. Authors may provide the signaling cartoon of YAP-TAZ

3. Please provide good quality IHC pictures

4. Overall there is no positive significance except for HER2+/ER- patients. However, the number of these patients are really low

5. Quality of provided figures are poor

6. Results from Machine Learning may be improved

7. All most 25-30% HER2+ BC have PIK3CA mutation. Is there any any correlation of activating PI3K signaling along with YAP-TAZ signaling? Any comments from authors side

8. Discussion is way too lengthy 

Minor comments:

Tables are not readable

Typo error in the table

Reviewer 2 Report

This works studied the clinical and prognosis impact of the hippo pathway in HER2+ breast cancers in

means of an immunohistochemical analysis.

The hippo pathways in breast cancer are an interesting ways for prognosis or targeting therapy.

MAJOR COMMENTS:

- In the method (line 185...) clone and manufacture of antibodies were described but not

dilution and retrieval procedure that are necessary for all immunohistochemical study. Also,

antibodies of new biological pathways could be confirmed by negative (biological negative

and technical negative) and positive controls (confirmed by a mRNA technic such as cell

cultures or confirmed biopsy).

- Only 2/65 patients were ER negative, so HER2 enriched. It could be interesting to compare

these 2 groups ER+HER+ and ER-HER2+ because Hippo pathways plays a rule in ER+ breast

cancers (YAP interact negatively with ER). 5 to 10 cases ER-HER+ were enough for statistical

analysis.

- Why don’t the authors study also Luminal HER2- and triple-negative breast cancers?

MINOR COMMENTS

- Precise the version (years) of ASCO-CAP guidelines (line 193 and...)

- Score is complex for a routine analysis: simplification could be interesting.

Reviewer 3 Report

This study has attempted to perform a IHC study on a small sample of 65 Her2+ breast tumors to identify a possible predictor of pCR after neoadjuvant treatment. IHC was performed with markers for the Hippo pathway, as a candidate influencer of treatment outcome. Expression of several markers correlated with the absence of pCR (or presence of residual disease), but this effect was lost when adjusted for ER-status and BMI.

Comments

1.        In a prior study by these authors, the correlation with high TAZ worked only in Her2+, why? And if it’s positively associated with pCR, why is it negatively associated with DSF? (p6, line 142)

2.        Methods should describe more clearly how initial tumor material was derived to select Her2+cases, including the failure rates to IHC-phenotype (p7, lines 160-167).

3.        This is a multicenter study, while Her2+ is found in 1 in 6 breast cancer cases; how come so few cases were included in this study?

4.        By its nature, the conventional core biopsy (14 gauge) is a small, rather limited, representation of the whole lesion. The limited nature of the sample which can also be fragmented and/or distorted, adds a layer of difficulty in making a definitive diagnosis compared with surgical excision specimens. Why no attempt to IHC-phenotype the tumors that did not show pCR and were removed surgically after neoadjuvant treatment?

5.        Have the authors attempted to replicate findings in the TCGA dataset? There is probably no data on pCR in TCGA, but there is data on survival.

6.        The HER2-Enriched biomarker identifies patients with a higher likelihood of achieving a pCR following neoadjuvant anti-HER2-based therapy beyond HR status and CT use (PMID: 32000054). Why wasn’t the same experiment performed with another group of tumors undergoing neoadjuvant treatment (but not Her2+)?

7.        The descriptive statistics in table 1 should be compared with an average group of Her2+ patients to rule out selection-biases. That data must be available from all the cases that were excluded from study-entry?

8.        The p-values for the correlations mentioned on p19 should be specified in the text. It is impossible to deduce from the figure.

9.        Likewise, it is not clear from which figure or table the correlations listed under par. 3.2 (p.20) are derived; p-values, odds ratios and confidence intervals are lacking.

10.     P20, line 418 refers to fig 2A, this must be a typo?

11.     The data presentation of fig 3A is problematic; the reader cannot make the comparison between pCR and non-pCR groups in this way, per marker. Also not clear what the p<2.2E-16 refers to. My impression is that fig 3B displays exactly the same as 3A, but then in a different line-up. If so, please omit fig 3A, or else explain the difference.

12.     P21, line 435 introduces logistic regression, but it is not clear from the text above which markers were “the clinical and molecular variables of interest”; this should be stated clearly with p-values, odds ratios, confidence intervals.

13.     The machine learning used all the biomarker expressions as raw values as “input independent variables”; but on p19 several biomarkers were shown to correlate. This should be clarified.

Minor note

The composite variable obtain by multiplying staining intensity and percentage of positive cells is said to be continuous (p9, line 214), but that’s impossible because the intensity is already categorical. For a true continuous variable, the intensity will need to be measured, e.g. by fluorescence or absorption.